# The Predictive Role of Hepatitis B Biomarkers on HBV Reactivation following Direct-Acting Antiviral Therapy in HBV/HCV Coinfected Patients

**DOI:** 10.3390/v14081812

**Published:** 2022-08-18

**Authors:** Chih-Wei Tseng, Wen-Chun Liu, Ping-Hung Ko, Yen-Chun Chen, Kuo-Chih Tseng, Ting-Tsung Chang

**Affiliations:** 1Department of Internal Medicine, Dalin Tzu Chi Hospital, Buddhist Tzu Chi Medical Foundation, Chiayi 62247, Taiwan; 2School of Medicine, Tzuchi University, Hualien 97004, Taiwan; 3Department of Internal Medicine, National Cheng Kung University Hospital, Tainan 70101, Taiwan; 4College of Medicine, National Cheng Kung University, Tainan 70101, Taiwan; 5Center of Infectious Disease and Signaling Research, National Cheng Kung University, Tainan 70101, Taiwan

**Keywords:** hepatitis B surface antigen, hepatitis B core-related antigen, HBV pregenomic RNA, hepatitis C, HBV/HCV coinfection, direct-acting antiviral, HBV reactivation

## Abstract

Hepatitis B and C (HBV/HCV) coinfected patients have a potential risk of hepatitis B reactivation (HBVr) after direct-acting antivirals (DAAs) treatment. The study intends to investigate the predictive role of HBV biomarkers in HBVr. Forty-six HBV/HCV coinfected patients receiving DAAs were enrolled. All patients completed treatment and follow-up to the 12th-week post-DAA treatment (P12). Blood samples were measured for HBV biomarkers, including hepatitis B surface antigen (HBsAg), hepatitis B core-related antigen (HBcrAg), and HBV pregenomic RNA (HBV pgRNA). The predictive factors for HBVr after DAA treatment were analyzed. Among 31 patients without nucleot(s)ide analogue (NA) treatment, seven (22.5%, 7/31) developed HBVr without hepatitis flare-up. Patients with HBVr had higher HBsAg titers than those without HBVr from baseline to P12 (*p* = 0.008, 0.009, 0.004, and 0.006 at baseline, week 4, end of treatment, and P12, respectively). The baseline HBsAg level was the only predictive factor associated with HBVr (HR, 2.303; 95% CI, 1.086–4.882; *p* = 0.030). In predicting HBVr, a baseline HBsAg titer > 20 IU/mL had a sensitivity, specificity, positive predictive value, and negative predictive value of 85.7%, 75.0%, 50%, and 94.7%, respectively. No patient had HBVr if the baseline HBsAg titer was <8 IU/mL. Serum HBcrAg and HBV pgRNA levels had no role in predicting HBVr. In conclusion, HBV/HCV coinfected patients are at risk of HBVr after DAA treatment. The baseline HBsAg level was the predictive factor associated with HBVr. Patients with a baseline HBsAg titer < 8 IU/mL can be considered as not having HBVr.

## 1. Introduction

Many patients with hepatitis C virus (HCV) have also been exposed to hepatitis B virus (HBV) in high-epidemic areas because they have similar transmission routes [1,2]. Patients with HBV/HCV coinfection have complex interactions between the two viruses [3,4] and, in most cases, HCV suppresses HBV [3,4]. When HCV is eradicated with direct-acting antivirals (DAAs), this suppressive effect of HCV is removed. HBV replication may increase, and HBV reactivation (HBVr) may occur [5,6,7]. However, a meta-analysis study demonstrated that only 24% of HBV/HCV coinfected patients who received DAA treatment had HBVr, and 9% developed HBVr-related hepatitis [8]. This means that not all patients develop HBVr after HCV eradication. Other immune mechanisms that can likely decrease or silence covalently closed circular DNA (cccDNA) transcription may be more active in patients without HBVr [9,10]. Recent evidence also supports that coadministration of entecavir during DAA treatment in HBV/HCV coinfected patients could prevent HBVr [7]. Hence, identification of the predictive factors for HBVr is an initial step in HBVr prevention.

Because the presence of HCV viremia may inhibit HBV DNA, HBV DNA is not the only factor to define the status of HBV infection [3,4]. Several HBV biomarkers such as hepatitis B surface antigen (HBsAg), hepatitis B core-related antigen (HBcrAg), and HBV pregenomic RNA (HBV pgRNA) are now available to reflect the activity of HBV and assist physicians in making clinical decisions [11,12,13]. These biomarkers also serve as surrogate markers in virally suppressed patients with undetectable HBV DNA under nucleos(t)ide analogue (NA) therapy [14,15,16,17]. A previous study from Taiwan demonstrated that HBsAg titer could be used to predict HBVr in HBV/HCV coinfected patients receiving DAAs [18]. The risk of HBVr is higher among patients with baseline HBsAg levels > 10 IU/mL [18]. However, the above evidence does not disclose the role of other HBV biomarkers (such as HBcrAg and HBV pgRNA).

This study aimed to investigate the predictive role of HBV biomarkers in HBVr. In addition, dynamic changes in HBV biomarkers in HBV/HCV coinfected patients after DAA treatment were also evaluated.

## 2. Patients and Methods

### 2.1. Patients

This prospective study included 46 patients with chronic HBV/HCV coinfection who received DAAs for chronic hepatitis C between May 2017 and July 2019 at the Dalin Tzu Chi Hospital, Buddhist Tzu Chi Medical Foundation, Chiayi, Taiwan. We included patients with (a) age > 20 years, (b) chronic hepatitis e antigen (HBeAg)-negative HBV infection (seropositive for HBsAg and seronegative for HBeAg > 6 months), and (c) chronic HCV infection with viremia (seropositive for HCV antibodies and detectable HCV RNA for more than 6 months). The exclusion criteria were autoimmune hepatitis, primary biliary cholangitis, sclerosing cholangitis, human immunodeficiency virus or hepatitis delta virus infection, overt hepatic failure, α1-antitrypsin deficiency, Wilson’s disease, or contraindications for DAA therapy.

A flow diagram of the patient allocation is shown in Figure 1. In this study, a total of 46 HBV/HCV coinfected patients receiving DAAs were included. All patients completed the DAA treatment and were followed up to 12 weeks post-DAA treatment. A total of 31 patients did not receive NA before and during DAA treatment (31/46; 67.4%).

### 2.2. Ethical Considerations

The study conformed to the ethical guidelines of the 1975 Declaration of Helsinki, as reflected by prior approval by the Ethics Committee of Dalin Tzu Chi Hospital (approval number B10904008). All patients signed informed consent forms before study commencement.

### 2.3. Clinical Monitoring

All patients received a scheduled follow-up at the gastrointestinal outpatient department. Blood samples were collected to evaluate the levels of HBsAg, HBcrAg, HBV pgRNA, HBV DNA, and HCV RNA at baseline, the 4th week during DAA treatment (W4), the end of DAA treatment (EOT), and the 12th-week post-DAA treatment (P12). In addition, serum HBeAg, HCV genotype, alanine aminotransferase (ALT), aspartate aminotransferase (AST), albumin, total bilirubin, estimated glomerular filtration rate (eGFR), prothrombin time, and platelet counts were analyzed.

HBVr was defined as (1) an increase in HBV DNA > 2 Log10 over that of baseline, (2) HBV DNA ≥ 100 IU/mL in a patient with previously undetectable level, or (3) an absolute level of HBV DNA > 20,000 IU/mL if baseline level was unavailable [19]. The high HBV DNA levels were defined as serum HBV DNA > 2000 IU/mL. Hepatitis flare-up was defined as an elevation of serum ALT level > 5 times the upper limit of normal and two times the baseline value [19]. Abdominal ultrasonography and alpha-fetoprotein (AFP) levels were evaluated at baseline. Clinical factors, including age, gender, and alcoholism, were recorded through chart review. Liver cirrhosis was diagnosed by radiologic cirrhosis or Fibrosis-4 (FIB-4) > 3.25 [20,21]. Radiologic cirrhosis was defined as coarse liver echotexture with nodularity and small liver size or the presence of features of portal hypertension (e.g., splenomegaly, ascites, and varices) based on liver imaging [22]. Hepatocellular carcinoma (HCC) was diagnosed either by biopsy or by imaging in the setting of liver cirrhosis. The specific imaging pattern is defined by increased contrast uptake in the arterial phase followed by contrast washout in the venous/delayed phase on computed tomography or magnetic resonance imaging [23]. A fatty liver diagnosis was based on results from abdominal ultrasound, including features of liver brightness, hepatorenal echogenicity contrast, deep attenuation, and vessel blurring [24].

### 2.4. HBV/HCV Quantification and HCV Genotyping

HBV DNA quantifications were performed using the COBAS^®^ HBV quantitative nucleic acid test on a COBAS^®^ 4800 System (Roche Diagnostics, Basel, Switzerland), with a lower limit of detection (LLOD) of 5 IU/mL. HCV RNA quantifications were measured using the COBAS^®^ AmpliPrep/COBAS^®^ TaqMan^®^ HCV Test v2.0 (Roche Diagnostics, Rotkreuz, Switzerland) with a LLOD of 15 IU/mL. HCV genotyping was performed using the COBAS^®^ HCV GT assay (Roche Diagnostics, Basel, Switzerland).

### 2.5. HBsAg Quantification

HBsAg quantification was measured using a two-step sandwich assay on a fully automated chemiluminescent enzyme immunoassay system (CLEIA) (Lumipulse G1200; Fujirebio, Inc., Tokyo, Japan) in a Lumipulse G1200 automated analyzer (Fujirebio, Inc.). HBsAg-HQ reagents were provided by Fujirebio Inc. The linear detection range of HBsAg-HQ was 5–150,000 mIU/mL. If the samples with HBsAg levels exceeded the upper detection limit, they were diluted 200-fold and retested.

### 2.6. HBcrAg Quantification

This HBcrAg assay was measured using a Lumipulse G1200 CLEIA analyzer (Fujirebio, Inc.) with a lower limit of detection of 2.0 Log U/mL and a linear range of 3.0–7.0 Log U/mL (1 kU/mL is equal to 3 Log U/mL). Samples with HBcrAg levels > 7 Log U/mL were diluted and remeasured. According to the Lumipulse^®^ G HBcrAg Immunoreaction cartridge product inserts, the accuracy of the test was confirmed by assay values (calculated with the assay values in Log U/mL) for three in-house controls ranging within ±5% of their control values.

### 2.7. Extraction and Reverse Transcription (RT) of HBV pgRNA

HBV RNA was extracted from 50 μL of serum using the Total RNA Extraction Miniprep System Kit according to the manufacturer’s instructions (Viogene, Taipei, Taiwan) and treated with DNase I (Thermo Fisher Scientific, Waltham, MA, USA). The isolated HBV RNA was reverse-transcribed using RevertAid reverse transcriptase (Thermo Fisher Scientific) with HBV-specific RT primers for HBV pgRNA. Before commencing RT, 20 μL of RNA, 1 μL of 10 μM primer, and 1 μL of 10 mM dNTPs (Thermo Fisher Scientific) were mixed, incubated at 70 °C for 5 min, and then placed immediately on ice for 1 min. RT was initiated by adding an RT reaction mix to a final volume of 35 μL at a final concentration of 1× RT buffer, 1 μL of RNAse inhibitor (Life Technologies, Carlsbad, CA, USA), and 1 μL of RevertAid reverse transcriptase (Thermo Fisher Scientific). The cycling conditions were as follows: 42 °C for 60 min followed by 75 °C for 5 min. Complementary DNA (cDNA) samples were stored at 4 °C before proceeding to quantitative real-time polymerase chain reaction (qPCR).

### 2.8. Quantification of Serum HBV pgRNA

Serum HBV pgRNA levels were detected by qPCR on a LightCycler 480 II Real-Time PCR Detection System (Roche, Mannheim, Germany) using the SYBR Green method. The primers were used to detect HBV 3.5 kb RNA [25]. The primers used to detect 3.5 kb HBV RNA are provided in Appendix A. The standards were constructed by PCR using each primer from the HBV full genome (accession number: KJ790199), and the PCR products were subsequently ligated into the T&A Cloning Vector (Yeastern Biotech, New Taipei City, Taiwan) [25]. The qPCR reaction mixture (20 μL) contained 10 μL 2× GoTaq^®^ Green Master Mix (Promega Corp., Madison, WI, USA), 0.5 μL forward primer (10 μM), 0.5 μL reverse primer (10 μM), 1 μL cDNA template, and 8 μL double distilled water. The reaction mixture was denatured at 95 °C for 5 min, followed by 40 cycles at 95 °C for 20 s and 60 °C for 40 s. The LLOD of serum HBV pgRNA was 1466 copies/mL, as calculated from probit analysis. The samples with HBV pgRNA levels below the LLOD were recorded as 1465 copies/mL (3.17 log copies/mL) for statistical analysis.

### 2.9. Statistical Analysis

Statistical analysis was performed using the SPSS Statistics version 19.0 software (SPSS Inc., Chicago, IL, USA). Categorical variables are presented as frequency counts and percentages of the total. The chi-square test or Fisher’s exact test was used to compare categorical data as appropriate. Continuous variables are expressed as medians and ranges. The Mann–Whitney *U* test was used to compare differences in continuous variables between the groups. In addition, single variable and multivariable logistic regression analyses were used to evaluate the factors associated with HBVr after DAA treatment. The multivariable logistic regression analyses with the stepwise elimination procedure were used to determine the factors associated with HBVr after DAA treatment, including those with a *p* value < 0.1 after single variable analysis. Receiver operating characteristic (ROC) analysis determined the most appropriate test cut-off value. All tests were 2-sided, and a *p*-value < 0.05 was considered statistically significant.

## 3. Results

### 3.1. Patient Characteristics

A total of 46 HBV/HCV coinfected patients, including 20 men and 26 women who received DAA treatment, were enrolled in this study. The baseline demographic, clinical, and virological characteristics of HBV/HCV coinfected patients at the initiation of DAA therapy are presented in Table 1. The median age was 65.9 years [range, 41–82 years]. Twenty-nine of the patients (63%) were infected with HCV genotype 1. The median HCV RNA level was 5.9 Log10 IU/mL. Nineteen (41.3%) patients had cirrhosis, including 18 Child–Pugh scores of A and one Child–Pugh score of B8. HCC was diagnosed in six (15.2%) patients at enrollment, including four following curative HCC therapy without viable tumor, and two with active HCC at initiation of DAA. Thirty-four (73.9%) patients received the sofosbuvir-based regimens, six (13.0%) received grazoprevir/elbasvir, five (10.9%) received paritaprevir/ritonavir/ombitasvir/dasabuvir, and one (2.2%) received glecaprevir/pibrentasvir. All 46 (100%) patients ultimately achieved a sustained virologic response at P12.

The median HBV DNA was 1.4 Log10 IU/mL with detectable HBV DNA in 25 (54.3%) patients (HBV DNA level: 2.3 [1.4–7.1] Log IU/mL) and high HBV DNA levels in five (10.9%) patients. A total of 15 (32.6%) patients received NA (all with entecavir) before DAA treatment, including 10 concurrently with DAA therapy and five prior to the initiation of DAAs. In patients receiving NA prior to the initiation of DAAs, the NA treatment started 16 months (range: 1–22) before the initiation of DAAs. For the patients receiving NA treatment, no one developed HBVr. Among 31 patients without NA before and during DAAs treatment (31/46; 67.4%), a total of seven (22.6%) patients developed HBVr. However, none of the patients had a hepatitis flare-up. Patients who received NA had significantly higher baseline serum HBV DNA, HBsAg, and HBcrAg levels than those without NA, while HBV pgRNA did not (Table 1). There were no statistically significant differences among the patients with/without NA regarding age, sex, fatty liver, alcoholism, HCCs, liver enzyme (ALT/AST), cirrhosis, FIB-4, AFP, albumin, total bilirubin, eGFR, prothrombin time, and platelet count.

### 3.2. HBV Biomarkers in Patients with and without NA Treatment

Figure 2 shows the dynamic changes in HBV biomarkers (HBV DNA, HBsAg, HBcrAg, and HBV pgRNA) in patients with and without NA treatment. For patients with NA treatment, HBV DNA was significantly suppressed from baseline to P12 (baseline vs. P12: *p* = 0.019, Figure 2A). The HBsAg, HBcrAg, and HBV pgRNA levels of the patients with NA treatment showed no significant differences between baseline and P12 (baseline vs. P12: *p* = 0.910, *p* = 0.195, and *p* = 0.426, respectively) (Figure 2B–D). For patients without NA treatment, the serum HBV DNA and HBcrAg levels increased from baseline to P12 (baseline vs. P12: *p* = 0.001 and *p* < 0.001, respectively, Figure 2A,C). The serum HBsAg and HBV pgRNA levels did not change from baseline to P12 (baseline vs. P12: *p* = 0.082 and *p* = 0.190, respectively; Figure 2B,D).

The baseline level of HBV DNA was higher in patients treated with NA (with NA vs. without NA, *p* < 0.001, Figure 2A), but the HBV DNA level decreased at P12 due to the effect of NA. Except for the serum HBsAg level at EOT, the patients with NA treatment had significantly higher HBsAg titers than those without NA treatment from baseline to P12 (baseline, *p* = 0.036; W4, *p* = 0.030; P12, *p* = 0.028, Figure 2B). HBcrAg and HBV pgRNA levels were not significantly different between patients with and without NA treatment from baseline to P12 (Figure 2C,D).

### 3.3. Subgroup Analysis in Patients without NA Treatment

The baseline demographic, clinical, and virological characteristics of the patients without NA treatment are presented in Table 2. Among 31 patients without NA treatment, seven patients (22.5%, 7/31) developed HBVr. Although two patients had liver cirrhosis, none developed hepatitis flare-up or hepatic failure after DAA treatment. The median ALT levels in patients with and without HBVr at P12 were 29 (range 13–44) IU/mL and 27 (range 12–72) IU/mL, respectively. Patients with HBVr were younger (*p =* 0.008) and had higher baseline HBsAg levels (*p =* 0.008) than those without HBVr.

Dynamic changes in HBV biomarkers (including HBV DNA, HBsAg, HBcrAg, and HBV pgRNA) in patients with and without HBVr are shown in Figure 3. In patients with HBVr, HBV DNA (*p =* 0.018) and HBcrAg (*p =* 0.026) levels significantly increased from baseline to P12 (Figure 3A,C). For patients without HBVr, serum HBcrAg and HBV pgRNA increased from baseline to P12 (*p* < 0.001 and *p* = 0.035, respectively) (Figure 3C,D). Patients with HBVr had significantly higher HBsAg titers than those without HBVr at baseline and during the follow-up period (*p* = 0.008, *p* = 0.009, *p* = 0.004, and *p* = 0.006 at baseline, W4, EOT, and P12, respectively) (Figure 3B). In addition, patients without HBVr had higher HBV pgRNA levels than those with HBVr (Figure 3D, *p* = 0.037).

### 3.4. Factors Associated with HBV Reactivation among the Patients without NAs Treatment

Table 3 shows the results of the single variable and multivariable logistic regression analyses. In single variable logistic regression analyses, baseline HBsAg level (hazard ratio [HR] 2.303; 95% CI 1.086–4.882; *p* = 0.030) and age > 65 years (HR, 0.069; 95% CI, 0.007–0.068; *p =* 0.022) were the factors associated with HBVr (Table 3). After adjustment for factors with *p* value < 0.1 in single variable analysis, baseline HBsAg level was the only predictive factor associated with HBVr in multivariable analyses with the stepwise elimination procedure (HR 2.303; 95% CI 1.086–4.882; *p =* 0.030).

Using receiver operating characteristic analysis, a cut-off value of 20 IU/mL was found to be the most appropriate baseline HBsAg titer for predicting HBVr in patients receiving DAA treatment without NA. In the prediction of HBVr, baseline HBsAg > 20 IU/mL had a sensitivity, specificity, positive predictive value, and negative predictive value of 85.7%, 75.0%, 50.0%, and 94.7%, respectively. All patients with HBVr had HBsAg levels of > 8 IU/mL. With a cut-off value of 8 IU/mL, the sensitivity and negative predictive values were 100%. The incidence of HBVr at P12 was 50.0% and 0% among patients with baseline HBsAg > 8 IU/mL and those < 8 IU/mL, respectively.

## 4. Discussion

The present study demonstrated that the HBV/HCV coinfected patients without NA treatment carried a risk of HBVr (22.5%) after DAA treatment but no patients with a hepatitis flare-up. Compared to baseline, the serum levels of HBV DNA and HBcrAg at P12 increased in patients without NA treatment. HBsAg levels were persistently higher in patients with HBVr than in those without HBVr from baseline to P12. In addition, the baseline HBsAg level with a cutoff value of 8 IU/mL could be used to exclude the risk of HBVr. Other HBV biomarkers (HBV DNA, HBcrAg, and HBV pgRNA) did not predict HBVr.

Our cohort demonstrated that HBV DNA levels increased after DAA treatment in patients without NA treatment (Figure 2A). Among the patients with HBV/HCV coinfection, HCV is usually the dominant virus that actively replicates and inhibits HBV replication [3,26]. Although the mechanism of this interaction is still partially understood, the direct or indirect (mediated by various host immune responses) viral interference has been proposed to explain the dominant role of HCV. In-vitro studies have demonstrated that the HCV “core” protein strongly inhibits HBV replication [27,28]. The HCV core protein can complex with HBV polymerase to prevent the binding of the polymerase protein to the HBV epsilon (ε) loop of pgRNA [3,27,28,29]. Another hypothesis states that HCV activates interferons that are responsible for the suppression of HBV [3,10,30]. Cheng et al. demonstrated that DAA-mediated clearance of HCV diminishes the stimulus for interferon secretion, further derepressing HBV replication [10]. The increase of HBV DNA could be possible in patients after the eradication of HCV with DAAs.

Despite the overall increase in HBV DNA levels, not all patients developed HBVr. The HBVr rate ranges from 2% to 57% [6]. The wide range of HBVr rates may be partly due to the differences in definitions of HBVr, baseline virological status, and different patient characteristics. A meta-analysis study involving 242 patients with chronic HBV infection demonstrated that the overall risk of HBVr was 24% (95% CI, 19–30) [8]. The reactivation rate was similar to our study (22.5%). The unpredictable and wide range of HBVr rates indicated the mechanisms by which baseline HCV core protein and interferons could suppress HBV, and these factors may be more active in patients without reactivation. Hence, identification of predictive factors for HBVr is important for prophylaxis treatment.

Another important endpoint is the risk of hepatitis flare-up. A previous meta-analysis study demonstrated that the risk of HBVr-related hepatitis was 9% (95% CI 5–16) [8]. The relative risk (RR) of HBVr-related hepatitis was significantly lower in patients with undetectable HBV DNA at baseline than in those with detectable HBV DNA (RR 0·17, 95% CI 0.06–0.50; *p* = 0·0011) [8]. In our cohort, the patients without NA treatment had a higher proportion of undetectable HBV DNA (61.3%) and a low HBV DNA level (0.7 Log IU/mL, range 0.7–3.2) at baseline. It is reasonable that no patient in our cohort developed a hepatitis flare-up. The median ALT level was as low as 29 IU/mL (range 13–44) at P12 among patients with HBVr. These data support that the incidence of hepatitis flare-up is rare in patients with a lower HBV viral load, although the risk of HBVr remains. This finding should be further explored in larger, well-defined groups to identify the optimal cut-off value of HBV DNA and enable informed decision-making.

The HBsAg titer has been used to predict HBVr in patients who discontinued NA therapy and to guide the cessation of antiviral therapy in HBeAg-negative CHB [13]. In the current study, the serum HBsAg level did not significantly change after HCV eradication in the overall and subgroups (with/without NA treatment or with/without HBVr). However, patients with HBVr had persistently higher HBsAg levels than those without HBVr from baseline to P12. This suggests that the baseline HBsAg titer reflects the possibility of HBVr. According to our analysis, baseline HBsAg was significantly associated with HBVr (HR, 2.303; 95% CI, 1.086–4.882; *p* = 0.030). A similar predictive value of HBsAg titer was reported in an observational study from Taiwan [18]. Yeh et al. showed that pretreatment HBsAg > 10 IU/mL was an independent factor associated with HBVr (HR, 2.80; 95% CI, 1.045–7.500; *p* = 0.041) and HBsAg seroclearance (HR, 0.328; 95% CI, 0.137–0.787; *p* = 0.012) [18]. Those findings suggest that patients with higher baseline HBsAg titers carry a higher risk of HBVr. On the other hand, our study also showed that the sensitivity and negative predictive value were 100% with an HBsAg cut-off value of 8 IU/mL. This finding could assist physicians in making decisions regarding NA prophylaxis in HBV/HCV coinfected patients while initiating DAAs.

Unlike serum HBsAg, serum HBcrAg and HBV pgRNA levels did not predict HBVr in our study. HBcrAg showed no change in patients with NA after DAA treatment (Figure 2C, *p =* 0195). On the other hand, the serum HBcrAg titer rises during DAA treatment in patients without NA treatment, no matter if the patients are with or without HBVr. HBcrAg is a biomarker comprising several antigens expressed from the precore/core gene: HBcAg, HBeAg, and 22-kD precore fragments [31]. Several reports have shown that serum levels of HBcrAg are closely correlated with intrahepatic cccDNA levels and reflect its transcriptional activity [14,16,31,32,33]. Our finding supports that serum HBcrAg could be a marker of the restoration of HBV replication after HCV eradication.

HBV pgRNA is reverse-transcribed to HBV DNA by the viral polymerase in the cytoplasm of hepatocytes before release of the enveloped virions [34]. Previous studies have indicated that the HCV core protein can complex with HBV polymerase to prevent the binding of the polymerase protein to the HBV epsilon (ε) loop of pgRNA [3,27,28,29]. Other studies hypothesize that interferons exert control over HBV replication via transcriptional suppression of cccDNA [3,10,30]. The HBV pgRNA level should increase during DAA treatment [10]. In our study, the serum HBV pgRNA was stable in most patients, and only the patients without HBVr had minor increases after DAA treatment. The correlation between HCV viremia and HBV pgRNA requires further study. In this study, the LLOD of the HBV pgRNA assay was 1466 copies/mL. It highlights the low sensitivity. Further test with lower cutoff values and higher sensitivity were required to clarify the role in prediction of HBVr.

This study has several limitations. First, there are around 3.4–12% of anti-HCV-positive patients who have coinfection of HBV in Taiwan [35,36]. The overall sample size is actually limited in this study due to the prevalence of HBV/HCV coinfection [8]. The number of HBVr cases was also small, owing to the nature of the reactivation rate. A small sample size may make it difficult to determine if a particular outcome is a true finding and, in some cases, a type II error may occur, i.e., the null hypothesis is incorrectly accepted and no difference between the study groups is reported. Further studies with a large sample size may be required to draw general conclusions about whether HBsAg or other biomarkers can be predictors of HBVr. However, this study was still the largest cohort that comprehensively investigated dynamic changes in various HBV biomarkers to our knowledge. This report also supports the role of HBsAg in predicting HBVr after multivariable analyses. Those findings could provide valuable information for clinical decisions. Second, patient selection bias should be considered. The criteria for the initiation of NA prophylaxis were based on discussions between the patients and doctors. Patients treated with NA had relatively high baseline HBV DNA levels. The real benefit of prophylaxis was not observed in this study. Because patients without NA treatment had lower baseline HBV DNA, the rate of hepatitis flare-up was low. Further studies using strict criteria for NA use are required to clarify this point. Third, only six cirrhotic patients received NA treatment in our cohort. The international guidelines recommend treating HBV cirrhosis to control HBV viremia, prevent the direct complications of the disease, and increase survival. However, the treatment of HBV in Taiwan was limited by the Bureau of National Health Insurance, which manages a single insurance fund and acts as a single payer in Taiwan’s health care market. Only cirrhotic patients with high viral load (>2000 IU/mL) could receive NAs by insurance, and all the patients with high viral load in our study received NAs treatment. Patient selection bias should be considered.

In conclusion, HBV/HCV coinfected patients are at risk of HBVr following DAA treatment. The baseline HBsAg level is a predictive factor for HBVr. HBcrAg and HBV pgRNA did not have any role in predicting HBVr. Patients with a baseline HBsAg titer < 8 IU/mL can be considered as not having HBVr and could assist physicians in making decisions regarding NA prophylaxis in HBV/HCV coinfected patients while initiating DAAs.

## Figures and Tables

**Figure 1 viruses-14-01812-f001:**
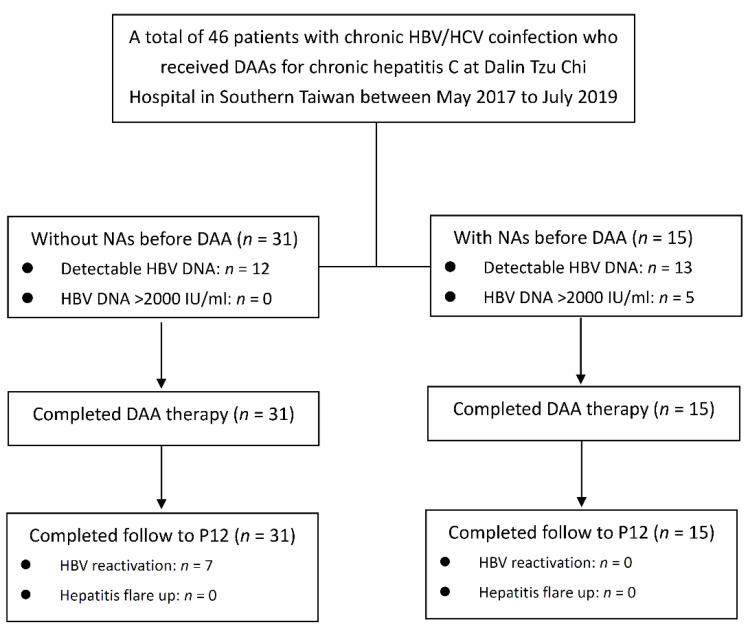
Patient allocation flow chart. DAA, direct-acting antiviral; NAs, nucleot(s)ide analogue; P12,12th week post-DAA treatment.

**Figure 2 viruses-14-01812-f002:**
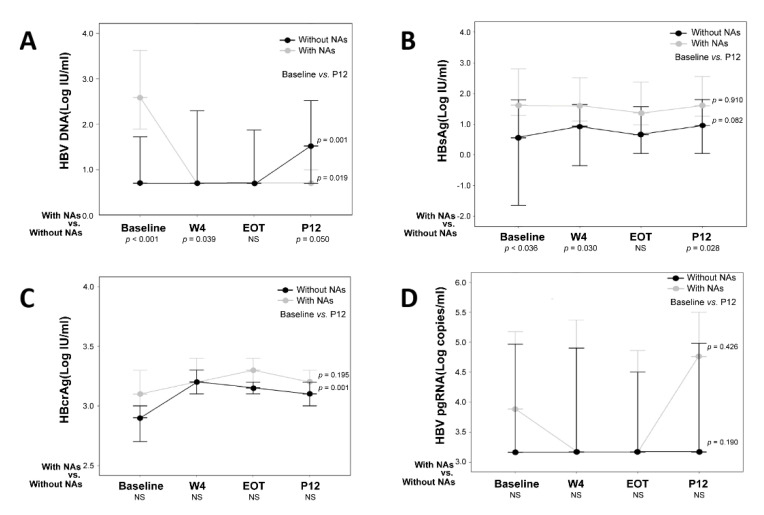
Expression of HBV DNA, HBsAg, HBcrAg, and HBV pgRNA in HBV/HCV coinfected patients with/without nucleot(s)ide analogue treatment (*n* = 46). (**A**) HBV DNA significantly decreased in patients with NAs treatment (*p* = 0.019) and increased in patients without NAs treatment (*p* = 0.001) from baseline to P12. (**B**) Serum HBsAg has no significant change from baseline to P12 in patients with/without NAs treatment. Except for the HBsAg titer at EOT, the patients with NA treatment have a significantly higher HBsAg titer than those without NA treatment. (**C**) Serum HBcrAg increased from baseline to P12 in the patients without NA treatment (*p* < 0.001). (**D**) Serum HBV pgRNA has no change from baseline to P12 in patients with and without NA treatment. HBV, hepatitis B virus; HCV, hepatitis C virus; HBsAg, hepatitis B surface antigen; HBcrAg, hepatitis B core-related antigen; pgRNA, pregenomic RNA; EOT, end of DAA treatment; P12, 12th week after DAA treatment; NAs, nucleot(s)ide analogue; NS, not statistically significant.

**Figure 3 viruses-14-01812-f003:**
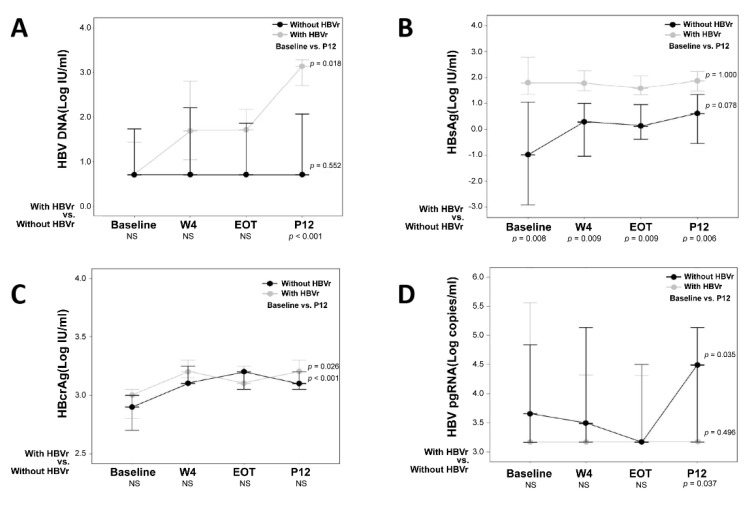
Expression of HBV DNA, HBsAg, HBcrAg, and HBV pgRNA with/without HBV reactivation in HBV/HCV coinfected patients without nucleot(s)ide analogue treatment (*n* = 31). (**A**) HBV DNA significantly increased from baseline to P12 in patients with HBV reactivation (*p* = 0.018). The HBV DNA titer in patients with HBV reactivation was higher than those without HBV reactivation at P12 (*p <* 0.001) (**B**) In patients with or without HBV reactivation, HBsAg has no significant change from baseline to P12. The patients with HBV reactivation had significantly higher HBsAg titers than those without HBV reactivation at baseline and during the follow-up course. (**C**) Serum HBcrAg increased from baseline to P12 in the patients with and without HBV reactivations (*p* = 0.026 and *p* < 0.001, respectively). There is no difference between patients with and without HBV reactivation. (**D**) Serum HBV pgRNA increased from baseline to P12 in the patients without HBV reactivation (*p* = 0.035). The patients without HBV reactivation had significantly higher HBV pgRNA than those with HBV reactivation at P12 (*p =* 0.037). HBV, hepatitis B virus; HCV, hepatitis C virus; HBsAg, hepatitis B surface antigen; HBcrAg, hepatitis B core-related antigen; pgRNA, pregenomic RNA; EOT, end of DAA treatment; P12, 12th week post-DAA treatment; NAs, nucleot(s)ide analogue; NS: not statistically significant.

**Table 1 viruses-14-01812-t001:** Baseline clinical and virological characteristics of the HBV/HCV coinfected patients receiving DAA therapy (*n* = 46).

	Total (*n* = 46)	Without NAs Treatment (*n* = 31)	With NAs Treatment (*n* = 15)	*p* Value
Age (years) ^&^	65.9 (41–82)	66.1 (41–82)	64.2 (49–81)	0.535 ^§^
Male (*n*, %)	20 (43.5)	11 (35.5)	9 (60.0)	0.116 ^†^
Cirrhosis (*n*, %)	19 (41.3)	13 (41.9)	6 (40.0)	0.901 ^†^
HCC (*n*, %)	6 (13.0)	3 (9.7)	3 (20.0)	0.375 ^‡^
Fatty liver (*n*, %)	15 (32.6)	11 (35.5)	4 (26.7)	0.740 ^‡^
Alcoholism (*n*, %)	5 (10.9)	4 (12.9)	1 (6.7)	>0.999 ^‡^
HCV RNA (Log IU/mL) ^&^	5.9 (3.7–7.2)	5.9 (3.7–7.2)	6.1 (4.9–7.0)	0.716 ^§^
Genotype (*n*, %)				0.381 ^†^
Type 1	29 (63.0)	18 (58.1)	11 (73.3)	
Type 2	14 (30.4)	10 (32.3)	4 (26.7)	
Type 6	3 (6.5)	3 (9.7)	0 (0)	
HBeAg-negative chronic infection (*n*, %)	41 (89.1)	31 (100)	10 (66.7)	0.002 ^†^
HBeAg-negative chronic hepatitis (*n*, %)	5.0 (10.9)	0 (0)	5 (33.3)	0.002 ^†^
HBV DNA (Log IU/mL)	1.4 (0.7–7.1)	0.7 (0.7–3.2)	2.6 (0.7–7.1)	<0.001 ^§^
Detectable HBV DNA (*n*, %)	25 (54.3)	12 (38.7)	13 (86.7)	0.002 ^†^
HBV DNA (Log IU/mL) in patients with detectable HBV DNA ^&^	2.3 (1.4–7.1)	2.2 (1.4–3.2)	2.9 (1.6–7.1)	0.030 ^§^
HBsAg (IU/mL) ^&^	13.2 (3 × 10^−4^–2347.7)	5.2 (3 × 10^−4^–2178.3)	35.0 (7 × 10^−4^–2347.7)	0.036 ^§^
HBsAg (Log IU/mL) ^&^	1.1 (−3.5–3.4)	0.7 (-3.5–3.3)	1.5 (-3.2–3.4)	0.036 ^§^
HBcrAg (Log IU/mL) ^&^	3.0 (2.5–6.5)	2.9 (2.5–3.6)	3.0 (2.7–6.5)	0.063 ^§^
HBV pgRNA (Log copies/mL) ^&^	3.3 (3.2–6.8)	3.2 (3.2–6.4)	3.9 (3.2–6.8)	0.523 ^§^
Detectable HBV pgRNA (*n*, %)	23 (50.0)	14 (45.2)	9 (60.0)	0.365 ^†^
HBV pgRNA (Log copies/mL) in patients with detectable HBV pgRNA ^&^	5.0 (3.4–6.8)	5.0 (4.1–6.4)	4.9 (3.4–6.8)	0.600 ^§^
FIB-4 ^&^	2.4 (0.5–12.3)	2.2 (0.5–12.3)	3.6 (1.3–6.3)	0.059 ^§^
Total bilirubin (mg/dL) ^&^	0.7 (0.3–1.7)	0.7 (0.3–1.7)	0.8 (0.5–1.2)	0.530 ^§^
ALT (U/L) ^&^	69 (25–281)	69.5 (25–235)	72 (25–281)	0.824 ^§^
AST (U/L) ^&^	47 (17–282)	47 (17–144)	58 (24–282)	0.566 ^§^
Albumin (g/dL) ^&^	4.4 (3.0–4.9)	4.3 (3.0–4.9)	4.4 (3.6–4.7)	0.598 ^§^
Prothrombin time (sec) ^&^	10.7 (9.7–12.4)	10.7 (9.7–12.4)	10.8 (10.0–12.2)	0.814 ^§^
AFP (U/L) ^&^	4.5 (1.3–1812.5)	4.2 (1.3–1812.5)	8.5 (1.8–75.5)	0.320 ^§^
Platelets (× 10^3^/mm^3^) ^&^	176.5 (56–316)	181 (56–316)	160 (80–210)	0.071 ^§^
eGFR (mL/min/1.73 m^2^) ^&^	83.7 (26.7–142.5)	87.3 (39–118.1)	72.5 (26.7–128.0)	0.598 ^§^
DAA regimen (*n*, %)				0.529 ^†^
Ledipasvir/sofosbuvir	27 (58.7)	18 (58.1)	9 (60.0)	
Daclatasvir + sofosbuvir	4 (8.7)	2 (6.5)	2 (13.3)	
Paritaprevir/ritonavir/ombitasvir + dasabuvir	5 (10.9)	4 (12.9)	1 (6.7)	
Elbasvir/grazoprevir	6 (13.0)	4 (12.9)	2 (13.3)	
Sofosbuvir + ribavirin	3 (6.5)	2 (6.5)	1 (6.7)	
Glecaprevir/pibrentasvir	1 (2.2)	1 (3.2)	0 (0)	

^&^ Data are expressed as median (range). ^†^ Chi-squared test; ^‡^ Fisher’s exact test; ^§^ Mann–Whitney U test HCV, hepatitis C virus; HBV, hepatitis B virus; HBsAg, hepatitis B surface antigen; HBcrAg, hepatitis B core-related antigen; pgRNA, pregenomic RNA; FIB-4, fibrosis-4 index; ALT, alanine aminotransferase; AST, aspartate aminotransferase; AFP, alpha-fetoprotein; eGFR, estimated glomerular filtration rate; NAs, nucleot(s)ide analogs; DAA, direct-acting antiviral.

**Table 2 viruses-14-01812-t002:** Baseline clinical and virological characteristics of HBV with/without reactivation in HBV/HCV coinfected patients receiving DAA therapy without nucleot(s)ide analogue treatment (*n* = 31).

	Without HBV Reactivation (*n* = 24)	With HBV Reactivation (*n* = 7)	*p* Value
Age (years) ^&^	67.9 (44–82)	58.3 (41–66)	0.008 ^§^
Age < 65 Y/O (*n*, %)	7 (29.2)	6 (85.7)	0.012 ^‡^
Male (*n*, %)	7 (29.2)	4 (57.1)	0.210 ^‡^
Cirrhosis (*n*, %)	11 (45.8)	2 (28.6)	0.667 ^‡^
HCC (*n*, %)	3 (12.5)	0 (0)	>0.999 ^‡^
Fatty liver (*n*, %)	7 (29.2)	4 (57.1)	0.216 ^‡^
Alcoholism (*n*, %)	2 (8.3)	2 (28.6)	0.212 ^‡^
Genotype 1 (*n*, %)	14 (58.3)	4 (57.1)	>0.999 ^‡^
HCV RNA (Log IU/mL) ^&^	6.0 (3.9–7.2)	5.7 (3.7–7.0)	0.139 ^§^
Sofosbuvir-containing Regimen (*n*, %)	17 (70.8)	5 (71.4)	>0.999 ^‡^
HBV DNA (Log IU/mL)	0.7 (0.7–3.2)	0.7 (0.7–2.4)	0.764 ^§^
Detectable HBV DNA (*n*, %)	10 (41.7)	2 (28.6)	0.535 ^‡^
HBV DNA (Log IU/mL) in patients with detectable HBV DNA ^&^	2.0 (1.4–3.2)	2.3 (2.2–2.4)	0.485 ^§^
HBsAg (IU/mL) ^&^	0.12 (3 × 10^−4^–1053.1)	62.7 (8.1–2178.3)	0.008 ^§^
HBsAg (Log IU/mL) ^&^	−0.9 (−3.5–3.0)	1.8 (0.9–3.3)	0.008 ^§^
HBcrAg (Log IU/mL) ^&^	2.9 (2.5–3.6)	3.0 (2.7–3.1)	0.661 ^§^
HBV pgRNA (Log copies/mL) ^&^	3.2 (3.2–6.4)	3.2 (3.2–6.2)	0.800 ^§^
Detectable HBV pgRNA (*n*, %)	11 (45.8)	3 (42.8)	>0.999 ^‡^
HBV pgRNA (Log copies/mL) in patients with detectable HBV pgRNA ^&^	4.8 (4.1–6.4)	6.1 (5.0–6.2)	0.225 ^§^
FIB-4 ^&^	2.2 (0.5–12.3)	1.4 (0.5–5.0)	0.317 ^§^
Total bilirubin (mg/dL) ^&^	0.7 (0.3–1.7)	0.6 (0.5–1.1)	0.473 ^§^
ALT (U/L) ^&^	64 (25–204)	73 (42–235)	0.695 ^§^
AST (U/L) ^&^	46 (17–144)	52 (17–141)	0.729 ^§^
Albumin (g/dL) ^&^	4.4 (3.0–4.9)	4.2 (3.8–4.6)	0.764 ^§^
Prothrombin time (sec) ^&^	10.7 (9.7–12.4)	10.7 (10.1–12.3)	0.872 ^§^
AFP (U/L) ^&^	4.8 (1.5–1812.5)	2.4 (1.3–67.3)	0.118 ^§^
Platelets (x 10^3^/mm^3^) ^&^	188 (56–316)	177 (87–243)	0.595 ^§^
eGFR (mL/min/1.73 m^2^) ^&^	75.8 (38.9–118.1)	91.8 (87.4–100.3)	0.139 ^§^

^&^ Data are expressed as median (range). ^‡^ Fisher’s exact test; ^§^ Mann–Whitney U test. HCV, hepatitis C virus; HBV, hepatitis B virus; HBsAg, hepatitis B surface antigen; HBcrAg, hepatitis B core-related antigen; pgRNA, pregenomic RNA; FIB-4, fibrosis-4 index; ALT, alanine aminotransferase; AST, aspartate aminotransferase; AFP, alpha-fetoprotein; eGFR, estimated glomerular filtration rate.

**Table 3 viruses-14-01812-t003:** Predictive factors for HBV reactivation in HBV/HCV coinfected patients receiving DAA therapy without nucleot(s)ide analogue treatment (*n* = 31).

	Single Variable Logistic Regression		Multivariable Logistic Regression	
	HR (95% CI)	*p* Value *	HR (95% CI)	*p* Value *
Age > 65 Y/O	0.069 (0.007–0.068)	0.022		
Male	3.218 (0.570–18.385)	0.185		
Cirrhosis	0.473 (0.076–2.935)	0.421		
Fatty liver	3.238 (0.570–18.385)	0.185		
Alcoholism	4.400 (0.494–39.210)	0.184		
Genotype 1	0.952 (0.174–5.228)	0.955		
HCV RNA (Log IU/mL)	0.570 (0.231–1.407)	0.223		
Sofosbuvir-containing regimen	1.029 (0.160–6.620)	0.976		
HBV DNA (Log IU/mL)	0.826 (0.255–2.676)	0.750		
HBsAg (Log IU/mL)	2.303 (1.086–4.882)	0.030	2.303 (1.086–4.882)	0.030
HBcrAg (Log IU/mL)	0.991 (0.034–28.628)	0.996		
HBV pgRNA (Log copies/mL)	1.195 (0.585–2.441)	0.625		
FIB-4	0.793 (0.485–1.299)	0.358		
Total bilirubin (mg/dL)	0.290 (0.011–7.507)	0.456		
ALT (U/L)	1.005 (0.990–1.019)	0.545		
AST (U/L)	0.996 (0.974–1.019)	0.743		
Albumin (g/dL)	1.014 (0.090–11.456)	0.991		
Prothrombin time (sec)	1.183 (0.363–3.857)	0.780		
AFP (U/L)	0.998 (0.988–1.009)	0.725		
Platelets (×10^3^/mm^3^)	0.996 (0.981–1.010)	0.579		
eGFR (mL/min/1.73 m^2^)	1.041 (0.988–1.098)	0.134		

* Statistics with a logistic regression model (single variable and multivariable analyses). HCV, hepatitis C virus; HBV, hepatitis B virus; HBsAg, hepatitis B surface antigen; HBcrAg, hepatitis B core-related antigen; pgRNA, pregenomic RNA; FIB-4, fibrosis-4 index; ALT, alanine aminotransferase; AST, aspartate aminotransferase; AFP, alpha-fetoprotein; eGFR, estimated glomerular filtration rate.

## Data Availability

The data of this study are available on reasonable request from the corresponding author.

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
