# Peer review of "The Predictive Role of Hepatitis B Biomarkers on HBV Reactivation following Direct-Acting Antiviral Therapy in HBV/HCV Coinfected Patients"

_viruses, 2022, doi:10.3390/v14081812_

Round 1
Reviewer 1 Report
Tseng and colleagues present a study trying to elucidate a predictor that could identify HBV/HCV coinfected patients that could be at risk of HBV reactivation following direct-acting anti-HCV therapy. The phenomenon that HCV usually suppress HBV is widely known. It was also shown in numerous studies that suppression of HCV by either PegIFN+/-ribavirin or direct-acting antivirals can lead to HBV reactivation. It was reviewed some time ago by Mavilia et al. in J Clin Transl Hepatol. 2018; 6(3): 296–305., who referenced 10 studies with direct-acting antivirals and 9 studies with PegIFN+/-ribavirin therapy that showed varying level of HBV reactivation. Certainly, it would be beneficial to have a predictor at hand that could identify patients at the risk of HBV reactivation. Authors are implicating baseline HBsAg as such a marker. It is somewhat a follow up study of Yeh at el J Hepatol 2020;73(1):62-71 who showed that HBV reactivation is higher in patients with higher baseline HBsAg. The aim of the authors was to investigate also other HBV biomarkers next to HBsAg. Unfortunately, this study suffers from very low patients’ number especially those seven who experienced HBV reactivation. Authors correctly list this shortcoming in between many other limitations, but the reality stays: it is impossible to draw general conclusions if HBsAg or other biomarkers can be predictors of HBV reactivation based on such small sample size. The suggestion to not offer HBV/HCV coinfected patients with HBsAg below 8 IU/ml NA prophylaxis is not sufficiently supported by presented data. Another eyebrow-raising issue in this study is the high number of cirrhotic patients not receiving NA treatment. To start NA prophylaxis in such patients is mandatory and excuse that NA prophylaxis initiation was based on discussion between the patients and doctors is weak.
Reviewer 2 Report
This is an interesting study addressing the issue of HBV reactivation in HCV-infected patients receiving DAA treatment. The study is well performed, however, there are some points to be better elucidated and clarified:
- Please, provide more insights on the assay used for the quantification of serum HBV-RNA, particularly regarding the primers and probes and their localization and the type of standard used (HBV-DNA or HBV-RNA). Furthermore, the LLOD of the assay 1466 copies/ml, highlighting low sensitivity. This should be discussed in the text also in relationship to the results obtained.
- In the results section lines 190-191, the authors should report the N (%) of patients with detectable serum HBV-DNA and should report the median (IQR) serum HBV-DNA of these patients with detectable serum HBV-DNA. These data should be reported separately in group of ETV-treated patients and in drug-naïve patients. Similarly in the table, median (IQR) levels of serum HBV-DNA should be reported in those with detectable serum HBV-DNA
- Like serum HBV-DNA, the authors should report in the group of drug-treated and drug-naïve patients, how many patients have a detectable HBV-RNA and the related median (IQR) values.
- In Table 1, please don’t report HBsAg levels as logs, but as absolute values.
- The authors should also specify if patients were in the HBeAg-negative chronic infection or in the HBeAg-negative chronic hepatitis phase.
- Please, provide more information on when entecavir was started.
- Which is the rate of SVR in patients receiving DAA? Please, report and verify if there is a correlation with the results obtained.
- I suggest to repeat the multivariable analysis including also patients receiving entecavir, in order to confirm the protective role of treatment in preventing HBV-R.
- In Table 3, the HR (95%CI) was the same in the uni- and multi-variable analysis. Is this correct? Please, verify. The authors found that In prediction of HBVr, baseline HBsAg > 20 IU/ml is the best predictor for HBV-R. This threshold is very low, implying that almost all HBeAg-negative patients are at this HBV-R, supporting the clinical relevance of this phenomenon.
Reviewer 3 Report
This is an longitudinal study of HBV reactivation in HBV/HCV co-infected persons treated with direct acting agents.
This a well-written and informative study. The study inclusion / exclusion criteria and lab methodologies are well described.
The largest concern is that the overall sample size is quite modest, and it is unclear if the study has sufficient power to evaluate the association of clinical / sociodemographic variables with the risk of reactivation.
HBV genotype should be included.
Line 242: liver (not live)
